# Group 2i Isochrysidales produce characteristic alkenones reflecting sea ice distribution

Karen Jiaxi Wang [1,2✉], Yongsong Huang [1,2✉], Markus Majaneva[3], Simon T. Belt [4], Sian Liao[2,5], Joseph Novak [1], Tyler R. Kartzinel [2,6], Timothy D. Herbert [1,2], Nora Richter[1,2,7] & Patricia Cabedo-Sanz[4]

Alkenones are biomarkers produced solely by algae in the order Isochrysidales that have been used to reconstruct sea surface temperature (SST) since the 1980s. However, alkenone-based SST reconstructions in the northern high latitude oceans show significant bias towards warmer temperatures in core-tops, diverge from other SST proxies in down core records, and are often accompanied by anomalously high relative abundance of the $C_{37}$ tetra-unsaturated methyl alkenone ($\%C_{37:4}$). Elevated $\%C_{37:4}$ is widely interpreted as an indicator of low sea surface salinity from polar water masses, but its biological source has thus far remained elusive. Here we identify a lineage of Isochrysidales that is responsible for elevated $C_{37:4}$ methyl alkenone in the northern high latitude oceans through next-generation sequencing and lab-culture experiments. This Isochrysidales lineage co-occurs widely with sea ice in marine environments and is distinct from other known marine alkenone-producers, namely *Emiliania huxleyi* and *Gephyrocapsa oceanica*. More importantly, the $\%C_{37:4}$ in seawater filtered particulate organic matter and surface sediments is significantly correlated with annual mean sea ice concentrations. In sediment cores from the Svalbard region, the $\%C_{37:4}$ concentration aligns with the Greenland temperature record and other qualitative regional sea ice records spanning the past 14 kyrs, reflecting sea ice concentrations quantitatively. Our findings imply that $\%C_{37:4}$ is a powerful proxy for reconstructing sea ice conditions in the high latitude oceans on thousand- and, potentially, on million-year timescales.

[1] Department of Earth, Environmental and Planetary Sciences, Brown University, Providence, RI 02912, USA. [2] Institute at Brown for Environment and Society, Brown University, Providence, RI 02912, USA. [3] Norwegian Institute for Nature Research (NINA), NO-7485 Trondheim, Norway. [4] Biogeochemistry Research Centre, School of Geography, Earth and Environmental Sciences, Plymouth University, Plymouth PL4 8AA, UK. [5] Department of Chemistry, Brown University, Providence, RI 02912, USA. [6] Department of Ecology and Evolutionary Biology, Brown University, Providence, RI 02912, USA. [7] Department of Marine Microbiology and Biogeochemistry, NIOZ Royal Netherlands Institute for Sea Research, Texel, The Netherlands. ✉email: karen_wang@brown.edu; yongsong_huang@brown.edu

Alkenones are among the best proxies for paleo sea surface temperature (SST) reconstructions, and their exceptional diagenetic stability has enabled the generation of palaeo-temperature records spanning tens of millions of years of Earth's history[1–3]. They are a class of $C_{35-42}$ methyl and ethyl ketones with two to four double bonds produced exclusively by the algae in the order Isochrysidales[1,4–8]: in ocean settings *Emiliania huxleyi* and the closely related *Gephyrocapsa oceanica* have been considered as the exclusive producers of di-unsaturated ($C_{37:2}$) and tri-unsaturated ($C_{37:3}$) methyl alkenones[1,4,5]. The unsaturation index of alkenones ($U_{37}^{K'}$), defined as $C_{37:2}/(C_{37:2} + C_{37:3})$, is positively correlated with temperature, as validated both by algal culture experiments and global core-top calibrations[9–13].

However, application of $U_{37}^{K'}$-SST calibrations in the northern high latitude oceans has encountered major difficulties. Variable warm bias in $U_{37}^{K'}$-based SST reconstructions is often accompanied by the occurrence of tetra-unsaturated alkenone ($C_{37:4}$), which is usually absent in mid-to-low latitude oceans and *E. huxleyi* cultures even under exceedingly low growth temperature[8]. This pattern is reported in surface sediment and seawater filtered particulate organic matter (POM) samples from the Nordic Seas and the Bering Sea[14–16], and downcore sediment records in Okhotsk Sea during glacial periods[17–19]. As a result, Rosell-Melé[14] concluded that $U_{37}^{K'}$ was unreliable when the percentage of $C_{37:4}$ among the total $C_{37}$ methyl alkenones (%$C_{37:4}$) is >5%. Yet, %$C_{37:4}$ greater than 5% is widely reported in sediment records in the northern high latitude oceans, thereby complicating SST reconstructions from these regions (Supplementary Fig. 1).

On the other hand, negative correlations between %$C_{37:4}$ and sea surface salinity (SSS) on a regional scale were observed[14,15,20], with elevated %$C_{37:4}$ often associated with fresher and colder water mass[21,22]. The mechanisms for these observations, however, are poorly defined, with previous studies suggesting unknown oceanographic parameters[14] and/or alkenone producers[22] as the possible explanations. The occurrence of miscellaneous compounds that co-elute with $C_{37:4}$ alkenone in sediment samples extracts adds an additional level of complication when interpreting %$C_{37:4}$ data[23]. Regardless of these problems, elevated %$C_{37:4}$ is widely applied as an indicator of fresher and colder polar water mass or decreased SSS (Supplementary Table 1). More recently, it has been applied as an indicator of sea ice margin based on co-occurrence with benthic foraminifera signal suggesting sea ice breakup[24]. The interpretation of %$C_{37:4}$ as a proxy for SSS stands in direct contrast to results from culture experiments, which show positive or no correlation between salinity and %$C_{37:4}$ in marine and brackish Isochrysidales species[25–27]. Such conflict suggests that elevated %$C_{37:4}$ is not a physiological response to decreased SSS, but rather that unidentified producers of $C_{37:4}$ occur in high latitude marine environments[14,22].

In this study, we used next-generation sequencing (NGS) to identify Isochrysidales in water column, sea ice, and sediment samples in northern high latitude marine environments. We show that Isochrysidales communities in subarctic regions are composed of a lineage that is distinct from *E. huxleyi* and any other described species. Culture of this lineage isolated from sea ice demonstrates that they are prolific producers of $C_{37:4}$. %$C_{37:4}$ in POM and surface sediments in northern high latitude oceans show strong correlation with annual mean sea ice concentrations, which is further corroborated by downcore sediment records from the Svalbard region spanning the past 14 kyr. Our results suggest that %$C_{37:4}$ has great potential as a paleo sea ice proxy, which may provide crucial information about sea ice distributions prior to satellite observations.

## Results and discussion

**Identifying widespread ice-related alkenone producers**. We sequenced the V4 region of the 18S rRNA gene from three surface sediment samples containing high %$C_{37:4}$ (59–68%; Supplementary Fig. 2, Supplementary Data 1) that were collected from Victoria Strait in the Canadian Arctic Archipelago where the sea ice regime is mainly governed by first-year ice. Four Isochrysidales amplicon sequence variants (ASVs) were recovered from those surface sediments (Supplementary Table 2). We also extracted 18S-V4 sequences of Isochrysidales available from NGS datasets from high latitude oceans (20 datasets, 1216 samples; Supplementary Data 2). We used a phylogenetic framework to compare these data with published Isochrysidales 18S-V4 data from environmental and culture studies[28] (Fig. 1).

We followed the Isochrysidales classification of Theroux et al.[29] based on habitat salinity ranges and phylogenetic divergence: Group 1 (freshwater species, uncultured), Group 2 (brackish species, e.g., *Isochrysis galbana*, *Ruttnera lamellosa*), and Group 3 (marine species, *E. huxleyi* and *G. oceanica*). Our phylogenetic analysis revealed close relationships between the majority of Isochrysidales sequences recovered from marine sites with perennial or extended seasonal sea ice and those of published sequences from lakes with seasonal or perennial ice cover (Figs. 1 and 2a). The sequences formed a monophyletic group with 80% bootstrap support, and placement of these sequences within Group 2 Isochrysidales is supported by 100% bootstrap support. We refer to this monophyletic clade as the Group 2 ice lineage Isochrysidales (Group 2i). Group 2i is distinct from any named Isochrysidales species, and potentially represents one or more new species. The sequences falling within Group 2i were detected in environments with a wide range of salinity, from freshwater to saline brine channels within sea ice, and in samples from sea ice cores, under ice seawater, and sediments from regions governed by multi- and first-year ice. These distributions suggest that Group 2i is euryhaline and well-adapted to variable Arctic marine environments. *E. huxleyi* was abundant in seawater samples that were collected during the summer after ice-melt in the Chukchi Sea and Barents Sea, and was not identified in any of the sea ice samples except one sample from the central Arctic. Particularly, seawater samples collected in the Fram Strait during sea ice melting season (June 2014)[30] where the western part of the Fram Strait was ice-covered and eastern part was ice-free showed *E. huxleyi* was present in both ice-covered and ice-free sites. However, Group 2i was only detected in seawater samples collected under ice cover. The DNA data suggest that the presence of Group 2i is tightly associated with sea ice.

**Culture experiments show Group 2i produce predominant $C_{37:4}$**. To confirm that Group 2i produces high %$C_{37:4}$, we cultured the unclassified Isochrysidales strain RCC5486 at 3 and 6 °C in f/2 medium under salinity of 15, 21, 26, 31, and 38 ppt. The RCC5486 strain falls within the Group 2i in our phylogeny and it is the most phylogenetically similar strain to the Isochrysidales found in Canadian Arctic Archipelago surface sediments that is available in pure culture (Fig. 1). Live cells of RCC5486 were isolated during the algal pre bloom and bloom-development phase in sea ice samples from Baffin Bay[31] (67.47°N, 63.78°W; May–July 2016). The alkenones produced by RCC5486 showed exceptionally high %$C_{37:4}$ (~80%) at both 3 and 6 °C (Supplementary Fig. 2), and salinity does not have any significant influence on %$C_{37:4}$ (Supplementary Fig. 3). This further validates that physiological response to decreased salinity would not result in increased $C_{37:4}$ production. Published DNA sequences that fall into Group 2i in our phylogenetic analyses were also observed in water and

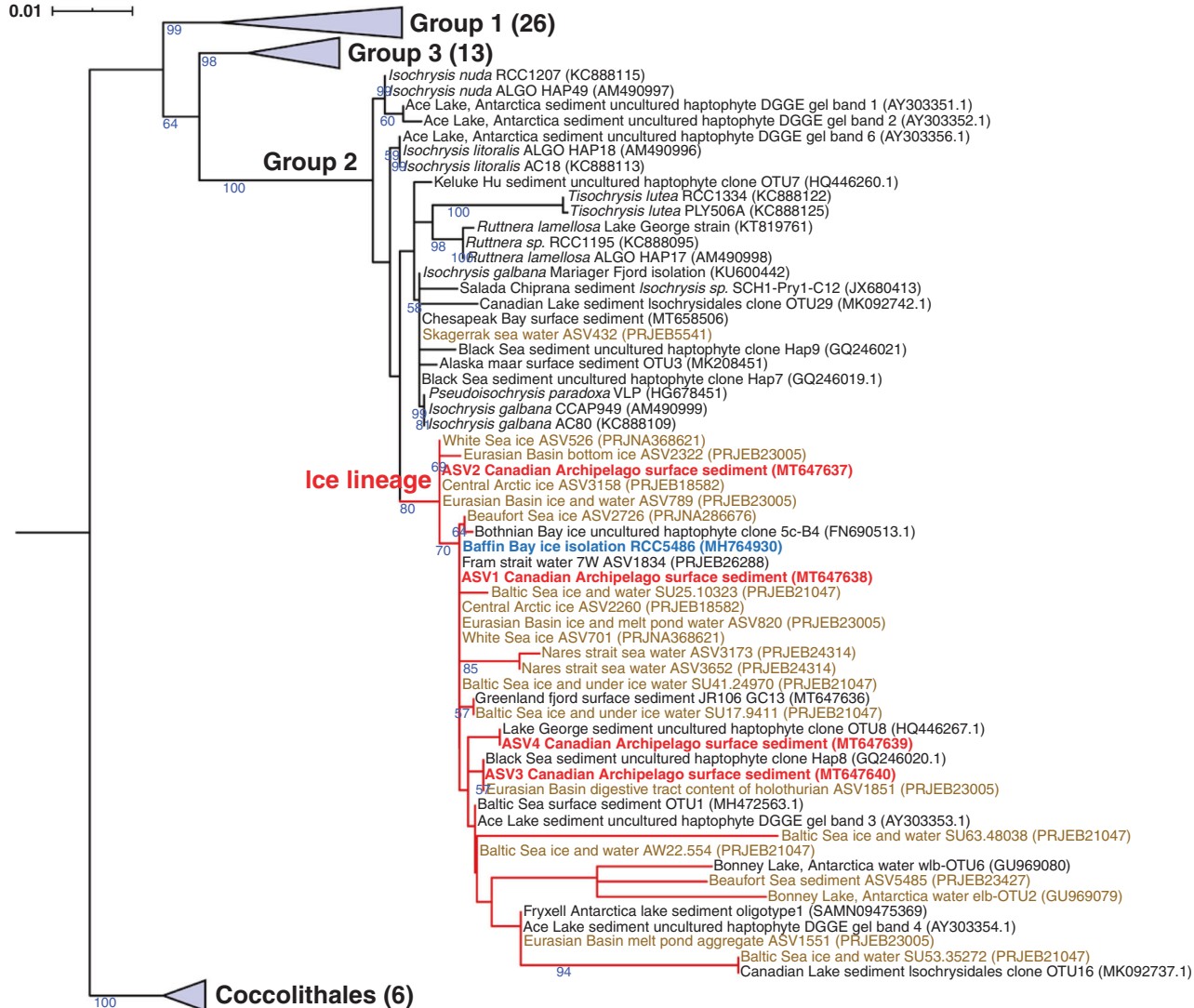

**Fig. 1 Phylogenetic position of the novel Group 2 ice lineage (Group 2i) Isochrysidales.** Maximum-likelihood tree based on 18S rRNA gene from 99 sequences spanning three groups of Isochrysidales and 6 from the Coccolithales outgroup. The numbers shown on branches represent bootstrap support for the node (only those >50% are shown). Numbers in the parentheses represent numbers of sequences included in the collapsed groups, and the expanded version of this tree can be found in Supplementary Fig. 4. The scale bar represents substitution/site. Sequences generated from Canadian Arctic Archipelago surface sediments in this study are highlighted in red, and sea ice isolated strain RCC5486 cultured in this study is highlighted in blue. Sequences from reanalyzed NGS datasets are highlighted in brown, and other sequences were obtained from NCBI GenBank. The branches of the ice lineage are marked in red.

sediment samples from lakes in China[29,32], Canada[33], the USA,[29] and Antarctica[34], where $C_{37:4}$ was the predominant alkenone. High %$C_{37:4}$ alkenone production is thus likely a shared characteristic across Group 2i Isochrysidales. Culture of RCC5486 also showed an absence of $C_{38}$ methyl alkenones ($C_{38}$Me), matching alkenone production by other Group 2 Isochrysidales and differing from *E. huxleyi*[8]. However, $C_{38}$Me was present in the Canadian Arctic Archipelago surface sediments, even though Group 2i was the only Isochrysidales detected through NGS. The $C_{38}$Me detected in Canadian Arctic Archipelago is likely produced by *E. huxleyi* even though their DNA was not detected. The discrepancy between DNA and alkenone preservation could result from differences in the life cycles of Group 2i compared to *E. huxleyi*: the former potentially forms cysts and enters a resting stage in surface sediment,[35,36] while the latter does not form resting spores during winter[37], after the bloom they die and disaggregate during downward transportation into sea floor[38].

**Implications for sea ice, SSS, and SST reconstructions.** The identification of the novel and widespread Group 2i Isochrysidales gives rise to the possibility of using %$C_{37:4}$ as a reliable sea ice proxy. Elevated %$C_{37:4}$ from surface sediment and seawater POM[15,22] is located at regions with seasonal sea ice in the Northern Hemisphere (Fig. 2b). %$C_{37:4}$ in both POM and surface sediment were positively and significantly correlated with sea ice concentrations (Fig. 3a, c). Differences in the calibration could be results of stronger seasonality of POM, uncertainty in the age of the surface sediment, and advection influencing the linkage between sediment and overlying surface water[12]. The positive correlation between sea ice concentration and %$C_{37:4}$ most likely reflect increasing alkenone contribution from Group 2i Isochrysidales with increasing sea ice. %$C_{37:4}$ showed negative and significant correlations with SSS in POM and surface sediments (Fig. 3b, d), consistent with prior research[14,15]. However, this relationship is a byproduct of the negative correlation between SSS and sea ice concentrations in the

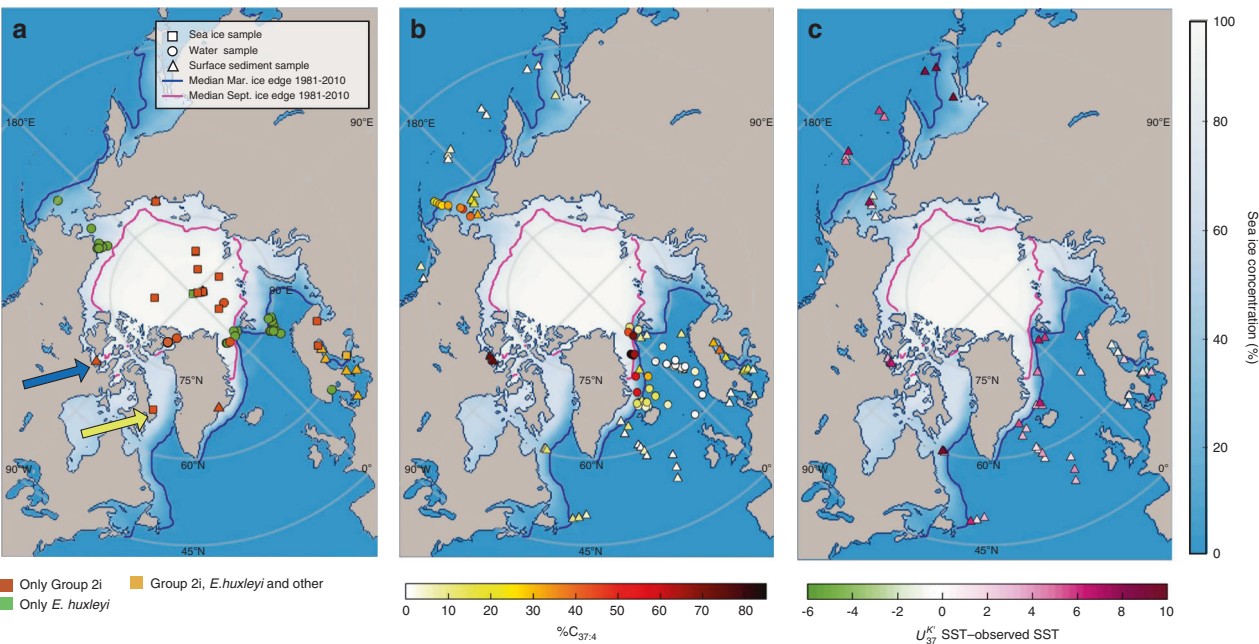

**Fig. 2 Distribution of the Group 2i Isochrysidales, %$C_{37:4}$ values, and $U^{K'}_{37}$-sea surface temperature bias. a** Distribution of Group 2i in northern high latitude marine environment from DNA sequencing in this study, reanalyzed NGS datasets, and NCBI GenBank based on phylogeny. Orange color represents samples with Group 2i detected, yellow color represents samples with both Group 2i and other Isochrysidales detected, and green color represents samples with occurrence of other Isochrysidales and absence of Group 2i (details of samples can be found in Supplementary Data 2). Blue arrow indicates surface sediment samples where both DNA sequencing and alkenone analyses were conducted in this study. Yellow arrow denotes the location where strain RCC5486 cultured in this study was isolated from. **b** %$C_{37:4}$ values of water filter POM compiled from Harada et al.[15] and Bendle and Rosell-Melé[16] are shown in circles. %$C_{37:4}$ values of surface sediment samples from this study are shown in triangles. **c** Differences between $U^{K'}_{37}$ reconstructed SST in surface sediment and 1981–2010 mean annual SST (WOA)[83]. The contour shows annual mean sea ice concentration during 1980–2010 from National Snow and Ice Data Center (NSIDC) Climate Data Record of Passive Microwave Sea Ice Concentration, Version 3, with a spatial resolution of 25 km × 25 km[84] (https://doi.org/10.7265/N59P2ZTG). The blue and magenta lines show 1980–2010 median sea ice extent during March and September, respectively, (NSIDC).

above sample sites, which reflects the temporal influence of melting sea ice on near-surface salinity (Supplementary Fig. 5). Thus variations in %$C_{37:4}$ cannot be attributed to SSS changes, and regional relationship between %$C_{37:4}$ and SSS should not be extrapolated for global-scale SSS reconstruction. So far there is no mechanistic explanation that supports a direct response of %$C_{37:4}$ to changes in SSS, except in low salinity (0–6 ppt) or estuarine environments where Group 1 Isochrysidales are present and contribute to elevated %$C_{37:4}$[29,39,40]. Group 1 Isochrysidales are typically found in freshwater to oligosaline environments and predominantly produce $C_{37:4}$[41]. Alkenone contribution from Group 1 can be identified through the presence of $C_{37:3b}$ isomer, which is not produced by Groups 2 and 3[41]. Our results show that both Group 1 and the Group 2i are present in brackish water underlying sea ice in the northwest of Gulf of Finland, Baltic Sea[42] (Supplementary Fig. 6). However, Group 1 Isochrysidales are very sparse within sea ice in the Baltic Sea and are absent in other sea ice samples examined in this study (Fig. 2). Thus, we conclude that the majority of $C_{37:4}$ production in the northern high latitude oceans distal from coast can be attributed to the presence of sea ice and Group 2i Isochrysidales.

Alkenones produced by the Group 2i are also a potential source of bias in $U^{K'}_{37}$–SST reconstructions. Surface sediment samples from regions covered by winter sea ice in the North Atlantic experience significant warm bias in $U^{K'}_{37}$–SST[13,43]. A similar warm bias is also observed in calculated SSTs from surface sediments examined in this study, including the Canadian Arctic Archipelago samples where Group 2i was detected (Fig. 2c); however, the exact mechanism leading to the warm bias is still a question. Group 2i may produce alkenones within multiyear sea ice, but

there is no prior report of alkenones detected from the surface sediments in the central Arctic. Bloom of Group 2i has been observed upon ice-melt in Lake George, USA, followed by a bloom of non-ice lineage Isochrysidales[44]. Group 2i and other Group 2 Isochrysidales are able to form cysts in sediments and potentially re-emerge to surface water during water-overturning under increased insolation[35,36], which could occur in blooms during ice-melt. Prolonged seasonal sea ice and the Group 2i Isochrysidales within sea ice could potentially shift the phenology of *E. huxleyi* from colder to warmer seasons. They are also likely to have a $U^{K'}_{37}$-SST calibration different from *E. huxleyi*. The bias in $U^{K'}_{37}$–SST introduced by Group 2i could thus be more prominent during periods with extended seasonal sea ice coverage in downcore records. More experimental and field studies are needed to fully understand the causes for the warm SST bias inferred by $U^{K'}_{37}$ in high latitude oceans.

**Validation of %$C_{37:4}$ as paleo sea ice proxy in downcore sediments.** Elevated %$C_{37:4}$ has long been observed in concordance with other sea ice proxies, such as IRD[45–49], diatoms[48,50], foraminifera assemblages,[24,51] and IP$_{25}$[52] in sediment records spanning the Late Pliocene to the Holocene. However, the interpretations of %$C_{37:4}$ were essentially derived from long-term SSS change instead of the occurrence of sea ice in previous studies (Supplementary Table 1).

To further evaluate %$C_{37:4}$ as a sea ice proxy within a paleoceanographic context, we compared %$C_{37:4}$ records with the widely applied sea ice proxy IP$_{25}$ and coarse fraction contents (% of grain size > 63 μm) in the Svalbard area during the past ca.

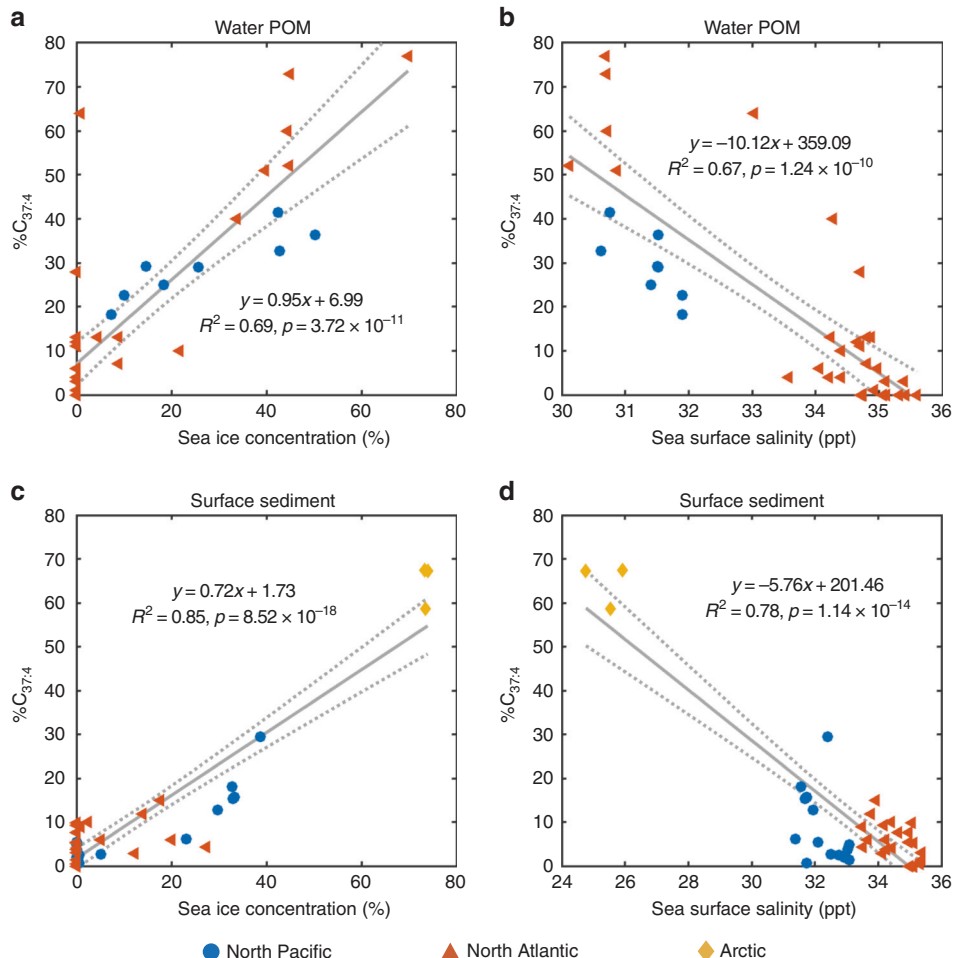

**Fig. 3 %C$_{37:4}$ showing significant positive correlation with annual mean sea ice concentration and negative correlation with sea surface salinity.**
**a** Linear regression between water filter POM %C$_{37:4}$ compiled from previous studies[15,16] and mean sea ice concentration 12 months prior to sample collection ($n = 40$ independent samples). **b** Linear regression between water filter POM %C$_{37:4}$ and measured SSS during collection[15,16] ($n = 40$ independent samples). **c** Linear regression between surface sediment %C$_{37:4}$ and 1980–2010 annual mean sea ice concentration ($n = 42$ independent samples). **d** Linear regression between surface sediment %C$_{37:4}$ and 1980–2010 annual mean SSS[85] ($n = 42$ independent samples). Surface sediment samples from Baltic Sea are not included in the regressions in **c** and **d**. Sea ice concentrations were obtained from NOAA/NSIDC Climate Data Record of Passive Microwave Sea Ice Concentration, Version 3, with a spatial resolution of 25 km × 25 km[84] (https://doi.org/10.7265/N59P2ZTG). The dashed lines denote 95% confidence interval for the regression.

14.5 kyr (Figs. 4 and 5). IP$_{25}$ is a C$_{25}$ isoprenoid lipid synthesized by certain Arctic sea ice diatoms and is deposited in underlying sediments following ice-melt in spring[53–55]. Coarse sediment fraction at this site is largely composed of lithic grains and foraminifera, reflecting sea ice formation off the western Barents shelf[56]. The Fram Strait is the major gateway between the North Atlantic and the Arctic Ocean, and sea ice distribution in our study area is highly sensitive to surface temperature variability in modern time[57]. Over longer timeframes, such as the Holocene, variations in sea ice extent in the Fram strait, the Greenland Sea, and the northern Barents Sea co-occurred with changes in Northern Hemisphere solar insolation and changes in Greenland air temperature[58–62]. Marine sediment core M23258-2[56,63] (74.99°N, 13.97°E) is located close to the modern maximum winter sea ice extent in the northeastern Norwegian Sea where temporal variations of sea ice conditions have been identified[64]. Alkenones were not detected prior to 14.1 kyr BP in M23258-2, likely due to seasonally extended sea ice cover limiting productivity. The general trend of %C$_{37:4}$ from M23258-2[63] is in direct agreement with inferred-temperature trends from the δ$^{18}$O values of Renland ice core located in east Greenland[65], with

increasing %C$_{37:4}$ corresponding to decreasing temperature, and is positively correlated with IP$_{25}$ concentration and coarse fraction content in M23258-2 (Supplementary Fig. 7). The high values of %C$_{37:4}$ in both M23258-2 and JM09-020-GC[52] (76.31°N, 19.70°E) located in the glacial trough of Storfjordrenna during the Bølling–Allerød (13.95–12.8 kyr BP) may represent partial sea ice cover and a retreating ice edge; high coarse fraction content during this time in M23258-2 suggests rapid breakup of the Svalbard–Barents Sea ice sheet. The highest peak in %C$_{37:4}$ in M23258-2 occurs at the beginning of the Younger Dryas (YD, ~12.8 kyr BP), indicating an expansion of sea ice. This is in agreement with increases in the coarse grain-size fraction and IP$_{25}$ from M23258-2 and increases in IP$_{25}$ at nearby site JM09-KA11-GC[66] (74.87°N, 16.48°E). The gradual decrease in %C$_{37:4}$ observed in M23258-2 and JM09-020-GC after the termination of YD (~11.7 kyr BP) is coincident with increasing Northern Hemisphere summer insolation and temperature. Low %C$_{37:4}$ in M23258-2 and the closely adjacent core SV04[67] (74.957°N, 13.899°E) suggests that sea ice extent retreated and was present in low concentrations at these sites between 10.8 and 4 kyr BP during the Holocene Thermal Maximum, while the fluctuating

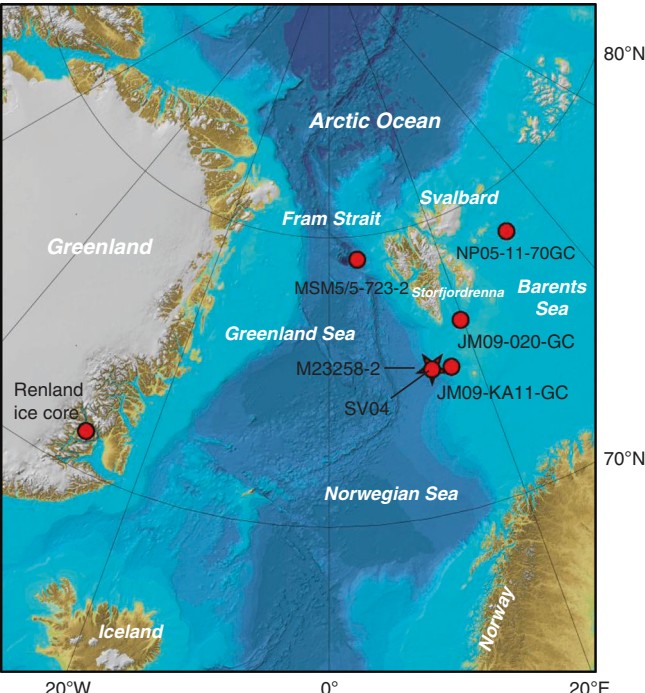

**Fig. 4 Map of the Svalbard region.** Red star[56,63] (this study) and red dots[52,60,64-68] are the locations of core sites cited in this paper. The bathymetry is adapted from IBCAO Version 3[86].

$\%C_{37:4}$ in JM09-020-GC suggests a variable ice-edge condition at Storfjordrenna during this period. $\%C_{37:4}$ in M23258-2 increased after ca. 4 kyr BP indicating a slight expansion of the sea ice margin during the neoglacial cooling period in the Northern Hemisphere, as the coarse fraction contents in this core also increased, reflecting sediment injections caused by cascades of dense brine water down the Barents shelf induced by seasonal sea ice[56]. $IP_{25}$ in M23258-2 is, however, absent (or below detection limit) during this time period. The return to partial seasonal sea ice cover during neoglaciation is also evident in the rapid $IP_{25}$ increase in the Fram Strait[60] (MSM5/5-723-2; 79.16°N, 5.33°E) and northern Barents Sea records[68] (NP05-11-70GC; 78.40°N, 32.42°E), as well as the re-emergence of $IP_{25}$ in JM09-KA11-GC.

In this study, we demonstrate that $\%C_{37:4}$ is a reliable and sensitive indicator for paleo sea ice, and has the potential for quantitative sea ice reconstructions based on surface sediment calibrations. As alkenones are solely produced by Isochrysidales, $\%C_{37:4}$ directly reflects proportion of Group 2i Isochrysidales within Isochrysidales community induced by sea ice, and is not affected by variations in total productivity or sedimentation rates as $IP_{25}$. $\%C_{37:4}$ is also less likely to be affected by diagenetic factors due to the strong diagenetic stability of alkenones, and does not suffer from dissolution like microfossil sea ice proxies such as diatoms, ostracods, and foraminifers. As an example of the durability of the $\%C_{37:4}$ proxy, a gradual increase of $\%C_{37:4}$ from 0 to 10% was observed in Bering Sea after the onset of MIS M2 glaciation (~3.3 Ma) in concordance with an increase in sea ice-related diatoms[50]. In addition, $\%C_{37:4}$ provides site-specific sea ice information which is different from proxies such as methanesulfonic acid and sea salt in ice from glaciers and ice sheets that records the averaged sea ice condition at a variable (e.g., depending on wind direction and speed) sector of ocean. Future work combining molecular biology and organic geochemistry analyses in culture experiments and large-scale analyses of seawater, sea ice, and sediment samples will improve the $\%C_{37:4}$-sea ice calibration and shed light on other sea ice properties such as seasonality.

## Methods

**Sample collection.** Surface sediment samples for both DNA sequencing and alkenone analyses were collected from Canadian Arctic Archipelago during Amundsen 2018. The samples were stored at −20 °C before DNA extraction. Other surface sediment samples for alkenone analyses were collected during several cruises including M86/1 (R/V Meteor; 2011), P435 (R/V Poseidon; 2012), EMB046 (R/V Elisabeth Mann Borgese; 2013), JR51 (James Clark Ross; 2000), JR106 (James Clark Ross; 2004), KN158-4 (Knorr; 1998), POS175 (Poseidon; 1990), HLY0601 (USCGC Healy; 2006), HLY0702 (USCGC Healy; 2007), M13/2 (Meteor; 1990), and M 21/2 (Meteor; 1992). The locations of the surface sediment samples are shown in Supplementary Data 1.

**NGS of Canadian Arctic Archipelago surface sediment samples.** Sequencing of the surface sediment samples was performed by Jonah Ventrues, Boulder, CO, USA, (https://jonahventures.com/). Genomic DNA was extracted from the samples using the DNeasy PowerSoil HTP 96, then a two-step PCR protocol was performed. First, a portion of 18S rRNA was amplified by an adapted version of primer pair 528Flong and PRYM01+7 (F: GCGGTAATTCCAGCTCCAA, R: GATCAGTGAAAACATCCCTGG, Egge et al.[69]). The first PCR step following the program described in Egge et al.[69] was performed in 25-μl reactions containing Phusion GC Buffer, 0.4 mM of each primer, 0.2 mM dNTP, DMSO 3%, 0.5 U Phusion polymerase (Thermo Scientific), and 2 μl of DNA template. After cleanup of the amplicons, a second round of PCR was performed to give each sample a unique 12-nucleotide index sequence. The final indexed amplicons were then cleaned and normalized using SequalPrep Normalization Kit (Life Technologies, Carlsbad, CA) and proceeded to sequencing on an Illumina MiSeq using V2 600-Cycle Kit (San Diego, CA). Sequencing success and read quality was verified using FastQC v0.11.8. The forward and reverse primers were removed from the sequences, and sequences with length below 100 bp were discarded. The ASVs were compiled for each sample after denoising by UNOISE3 algorithm with an alpha value of 5 to remove sequencing error[70]. Taxonomy of the ASVs was assigned using the SILVA reference database (including 506 Isochrysidales sequences) and assign_taxonomy.py as implemented in QIIME/1.9.1[71].

**Reanalyses of environmental sequences and phylogeny.** We searched for short-read studies including the term "sea ice" on the European Nucleotide Archive, and runs from studies targeting subfragments of 18S rRNA that included sea ice or seawater samples were downloaded. Primers were removed from the downloaded raw reads using cutadapt[72], except from Roche 454 sequenced reads where forward primers were removed after quality trimming. ASVs were compiled after the reads were quality trimmed using DADA2[73], except for studies with available quality trimmed ASVs[42,74]. The DADA2 parameters were adjusted based on the quality profiles of the sequencing runs. Taxonomic affiliations of the ASVs were identified using the assignTaxonomy command in DADA2 and, as reference sequences, the Protist Ribosomal Reference database[75] (including 159 expert-curated Isochrysidales sequences from each taxonomic groups). Sequences classified to the order Isochrysidales were selected to proceed to further analysis. Close relatives of the environmental sequences were identified through BLAST search against NCBI GenBank and compiled as reference sequences for phylogeny. All sequences were then aligned against a curated alignment from Gran-Stadniczeñko et al.[76] using PyNAST[77]. In total, the alignment contains 105 sequences, including 6 Coccolithales, 16 Groups 2 and 3 Isochrysidales cultures, 46 published environmental sequences from GenBank, and 37 recovered or reanalyzed environmental sequences from this study. A maximum-likelihood phylogeny was inferred using the RAxML[78] program with substitution model GTRCAT with 1000 bootstraps through CIPRES[79].

**Culture experiments.** We obtained the RCC5486 strain (GenBank accession: MH764930) from the Roscoff Culture from the National Institute for Environmental Studies. The strain was acclimatized to 3 and 6 °C for 10 days before the start of corresponding culture experiments. The f/2 medium was prepared from filtered and sterilized seawater collected from Vineyard Sound, Woods Hole, MA, USA, adjusted to 15, 21, 26, 31, and 38 ppt. Cultures were grown under a light:dark cycle set at 16:8 h. The light intensity was 140 μE · m$^{-2}$ · s$^{-1}$. Culture experiments under each salinity and temperature condition were triplicated.

Cultures were harvested at early stationary phase (monitored using hemocytometer counts (Hausser Scientific, PA, USA)) by filtering onto 0.7 μm glass fiber filters (Merck Millipore, MA, USA). All filters were wrapped with aluminum foils and immediately frozen at −20 °C before further extraction and analysis. For alkenones analysis, all of the filters were freeze-dried overnight and then dissolved in dichloromethane (DCM) and sonicated (3 × 30 min, 40 ml each time). The total extracts were separated and analyzed for alkenones as described in the section "Alkenones analyses."

**Alkenones analyses.** Surface sediment samples were analyzed for lipids by extracting around 10 g of freeze-dried sediment using a Dionex™ accelerated solvent extraction system with DCM:methanol mixture (9:1, v/v) to obtain a total lipid extract (TLE). The TLE was then separated into alkane, ketone, and polar fractions through silica gel columns by eluents with increasing polarity: hexane,

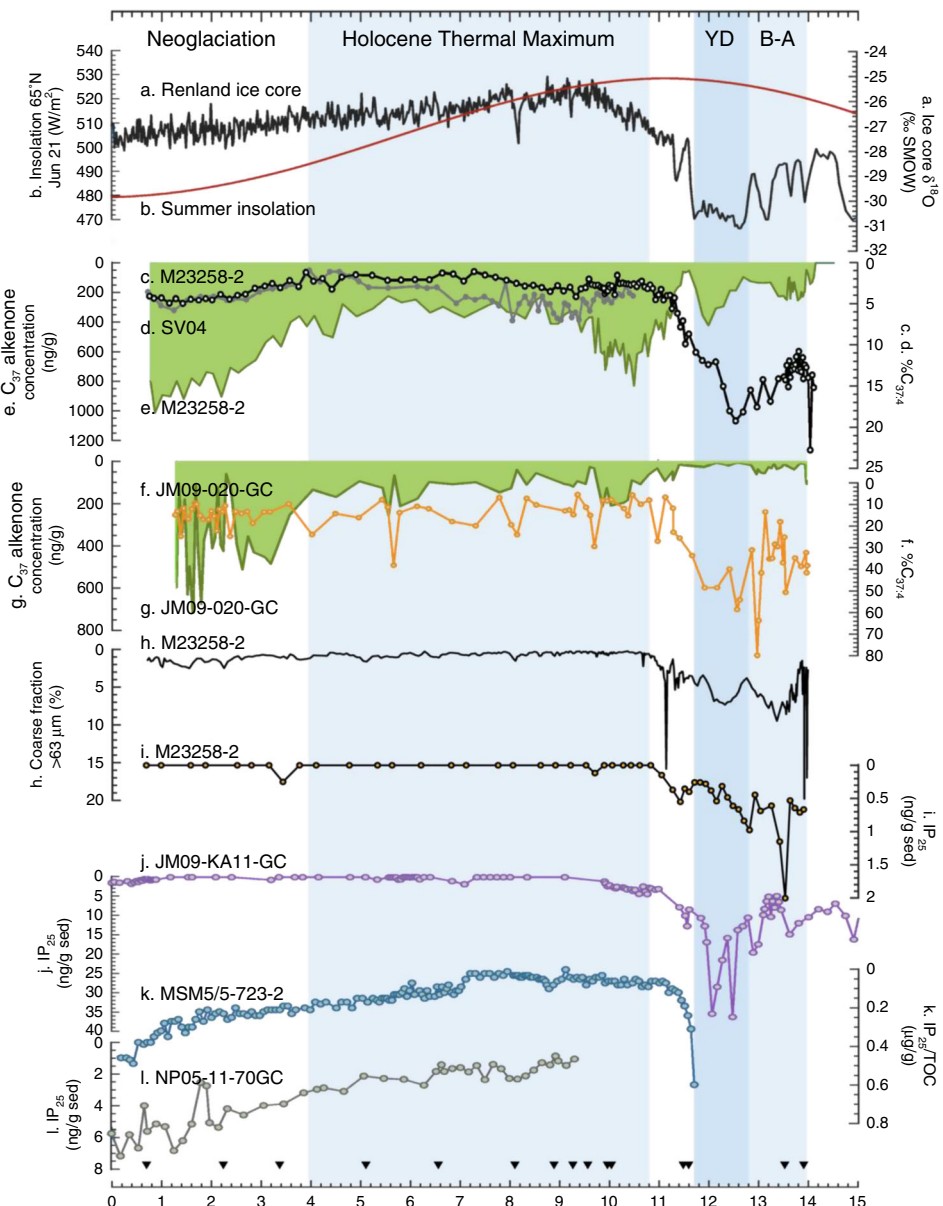

**Fig. 5 Sea ice reconstructions at the Svalbard region during the past ca. 14.5 kyr. a** δ[18]O from Renland ice core[65]. **b** Mid-July 65°N insolation[87].
**c** %$C_{37:4}$ from M23258-2[63]. **d** %$C_{37:4}$ from SV04[67]. **e** $C_{37}$ methyl alkenone concentration from M23258-2[63]. **f** %$C_{37:4}$ from JM09-020-GC[52]. **g** $C_{37}$ methyl alkenone concentration from JM09-020-GC[52]. **h** Coarse fraction from M23258-2[56]. **i–l** IP$_{25}$ concentration from M23258-2 (this study), JM09-KA11-GC[64,66], MSM5/5-723-2,[60] and NP05-11-70GC[68]. Black triangles represent age control points in M23258-2[56].

DCM, and methanol. The ketone fraction containing alkenones was analyzed by gas chromatography–flame ionization detection Agilent 7890N Series instrument equipped with mid-polarity column RTX-200 (105 m × 250 μm × 0.25 μm) with the oven program described in Zheng et al.[80]. This method is able to fully eliminate the co-elution between alkenones and alkenoates, as well as co-elution between $C_{38}$ methyl and ethyl alkenones[80]. Alkenone peaks were identified and quantified based on retention time compared with standard Group 1 alkenone samples extracted from BrayaSø Lake (Greenland) through Agilent ChemStation B.03.02. A gas chromatography–mass spectrometer (GC–MS, Agilent 7890B interfaced to 5977 inert plus MSD) equipped with RTX-200 columns (105 m × 250 μm × 0.25 μm) was used to examine co-elution. Samples with compounds co-eluting with alkenones were purified using silver thiolate silica gel columns with hexane: DCM (1:1, v/v), DCM, and acetone[23]. Regressions of alkenone indices and environmental parameters were performed via MATLAB (R2018b).

**IP$_{25}$ analyses.** Analysis of IP$_{25}$ in core M23258-2 was carried out as described by Belt et al.[81] with minor modification. An internal standard (9-octylhexadec-8-ene; 9-OHD; 0.1 μg) was added to dried sediment material (ca. 2 g), which was extracted using DCM/methanol (MeOH) (2:1 v/v; 3 × 6 ml) and ultrasonication. Following evaporation of the total organic extract (TOE) to dryness (N$_2$ stream; room

temperature), elemental S was removed according to Cabedo-Sanz and Belt[82]. Thus, the TOE was dissolved in hexane (1 ml) followed by addition of tetra-butylammonium sulfite (1 ml) and propan-2-ol (2 ml). After brief shaking by hand (1 min), ultrahigh purity water (3 ml) was added, reagitated (1 min), and centrifuged (2 min; 2500 rpm). The hexane layer was transferred to a clean vial and the procedure repeated twice more. Combined hexane extracts were dried using N$_2$. Partially purified TOEs were resuspended in hexane and purified using column chromatography (SiO$_2$; ca. 0.5 g in a glass pipette), with fractions containing IP$_{25}$ collected using hexane (6 ml). Final purification was achieved using Ag-ion chromatography (Supelco Discovery® Ag-Ion; 0.1 g). After elution of saturated hydrocarbons (hexane; 1 ml), unsaturated hydrocarbons including IP$_{25}$ were collected in acetone (2 ml).

Identification and quantification of IP$_{25}$ in purified sediment extracts was carried out via GC–MS using established methods[81] with an Agilent 7890 gas chromatograph equipped with a HP$_{5MS}$ fused-silica column (30 m; 0.25 μm film thickness; 0.25 mm internal diameter) coupled to an Agilent 5975 series mass spectrometric detector. IP$_{25}$ was identified by its characteristic retention index (RI$_{HP5-MS}$ = 2081) and mass spectrum. Quantification was carried out in single-ion monitoring mode. The resulting peak areas for IP$_{25}$ were normalized to internal standard responses, instrumental response factors, and sediment mass.

**Reporting summary**. Further information on research design is available in the Nature Research Reporting Summary linked to this article.

## Data availability
Raw sequence data generated in this study were archived in NCBI GenBank with accession number PRJNA650544, and assembled Isochrysidales sequences were archived in NCBI GenBank with accession number MT647637-MT647640. Reanalyzed NGS data can be found through European Nucleotide Archive with accession codes provided in Supplementary Data 2. Reference database SILVA is available at https://www.arb-silva.de/, and PR2 is available at https://github.com/pr2database/pr2database.

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

## Acknowledgements

This work was supported by the United States National Science Foundation (NSF) under grant EAR-1762431 and an IBES seed grant to Y.H., IBES Graduate Research, Training & Travel Award to K.J.W., and NSF DEB-1930820 to T.R.K. We would like to thank Dr. Beth Caissie for surface sediment samples from Bering Sea, and Dr. Daniel Sigman, Dr. Cara Manning, Dr. Anissa Merzouk, Dr. Alexandre Forest, Dr. Camille Wilhelmy, Dr. Philippe-Olivier Dumais, Dr. Diana Saltymakova, Dr. Katarzyna Polcwiartek, and Dr. Cindy Grant for surface sediment samples from Amundsen cruise. We would like to thank GEOMAR Helmholtz Centre for Ocean Research, BOSCORF repository, and Lamont-Doherty Core Repository, Columbia University, for other surface sediment samples. We would also like to thank Dr. Robert Spielhagen for sediment samples from core M23258-2.

## Author contributions

Y.H. initiated the research idea. K.J.W. and Y.H. codesigned the study after discussions with T.R.K., T.D.H., and S.T.B. Authors K.J.W., M.M., T.R.K., and N.R. analyzed the DNA and the phylogenetic data. S.L. conducted the culture experiment. J.N., S.L., K.J.W., Y.H., and T.D.H. conducted the alkenone analyses. S.T.B. and P.C.-S. conducted the IP25 analyses. K.J.W. and Y.H. cowrote the manuscript with contributions from coauthors.

## Competing interests

The authors declare no competing interests.
