## [Peer Review File · Nature Communications]

REVIEWERS' COMMENTS

Reviewer #1 (Remarks to the Author):

What are the noteworthy results?

Wang et al., produce new DNA, culturing and down-core evidence which resolves a long-standing issue in palaeoceanography. Specifically they find evidence that a distinct algal lineage is responsible for the high %C37:4 values observed in POM and sediment samples from sites influenced by Arctic water masses and propose this parameter can be used as a sea-ice proxy.

- Will the work be of significance to the field and related fields? How does it compare to the established literature? If the work is not original, please provide relevant references.
The work is within the context of a well-established suite of biomarkers – alkenones. It's rare for a major step forward in such circumstances, but because of the combination of culturing, next-gen sequencing and new sediment core analyses the authors have achieved this.

- Does the work support the conclusions and claims, or is additional evidence needed?

Yes

- Are there any flaws in the data analysis, interpretation and conclusions? - Do these prohibit publication or require revision?

No – it's a very thorough piece of work.

- Is the methodology sound? Does the work meet the expected standards in your field?

Yes

- Is there enough detail provided in the methods for the work to be reproduced?

Yes

Further comments:

The authors are to be commended on a very thorough piece of work. My only critique is that in the set-up and introduction of the paper they somewhat overstate that the community was interpreting %C37:4 as a SSS proxy and that the community had overlooked the possibility that elevated %C37:4 was due to an unidentified producer. In fact workers already suspected this and said as much. For example,

Even as early as 1998, Rosell-Mele was stating "Such large variability in 37:4% must relate to distinct environmental conditions between both regions, in relation to a different 37:4% vs SST relationship perhaps affected by an additional oceanographic parameter and/or interhemispheric species/strains variability."

And more clearly Bendle et al. 2005 (Wang et. al's ref 22) rejected %C37:4 as a SSS proxy and explicitly stated:

"Our results show that percent C37:4 can be used to reconstruct the relative extension of arctic/polar water masses in the North Atlantic...the results prevent confirmation of percent C37:4 as a paleo-SSS... The fact that *E. huxleyi* is rare in polar waters and *Gephyrocapsa* species are absent in those waters suggests that in this region there might be unknown alkenone producers associated with polar waters."

This interpretation was used by McClymont et al., 2008 and others who were careful not to interpret %C37:4 as a SSS proxy.

The authors should amend their introduction and discussion to reflect this prior thinking. For example at line 81-83 they state "Such conflict suggests that elevated %C37:4 is not a physiological response to decreased SSS, but rather that unidentified producers of C37:4 occur in high latitude marine environments".

Without citing any previous work (examples above) they give the reader the misleading impression that they are the first workers to propose this idea. They are not, but they are the first to prove it (for which they are commended)! They need to make a clear distinction in the introduction and discussion that are proving an idea that was already proposed by others.

Table S1: Need to make clear that these papers refer to down-core records (and not POM). Change from "List of published $C_{37:4}$ records..." to "List of published $C_{37:4}$ sediment-core records..."

Reviewer #2 (Remarks to the Author):

This is a well written and well executed study. The discovery of an extensive new lineage of $C_{37:4}$ alkenone producing group II isochrysidales and the discovery that $C_{37:4}$ can be used as a proxy to reconstruct sea ice extension in high latitude oceans for possibly millions of years is well worth publishing in Nature Communications. The study focused on the Arctic and it remains to be seen if the proxy also work for Antarctic environments. Namely, the newly discovered lineage is distantly related to the $C_{37:4}$ mK producers previously recovered from Ace Lake in the Vestfold Hills, Antarctica. These sequences were included in the phylogenetic tree but are clearly clustering in a separate group.

I only found a minor issue in the wording of the methods between lines 280-284. The sentence in line 280 ", except studies that quality trimmed ASVs were available" does not run well. IN line 282 the word "the" is missing after "using". The choice for the stand-alone sentence starting with "Sequences classified to the order... in line 284 is a bit odd. I am not sure what this refers to in the context that it is written.

Reviewer #3 (Remarks to the Author):

The study of polar climate change is a very timely topic given the vulnerability of polar regions to anthropogenic change. A new tool for assessing the past coverage of sea ice, proposed here, is especially key because climate models increasingly invoke sea ice as a major amplifier of climate perturbations from volcanic cooling to shutdown of Atlantic meridional overturning circulation. Therefore the paper is likely to be of very high interest to a broad research community, and is likely to catalyze both new studies and a wide re-interpretation of many existing datasets.

The article provides an important and compelling new interpretation for the contribution of an unusual lipid biomarker in sediments and thereby a new perspective on sea ice and polar climate change. For decades the concentration of $C_{37:4}$ alkenone has been observed to increase at certain periods in North Atlantic cores, but the environmental meaning of this change has been much disputed, with interpretations ranging from extreme cold to extreme low salinity. This study convincingly shows through laboratory culture experiments isolating a $C_{37:4}$ producing strain, that salinity does not affect the $C_{37:4}$ ratio of the produced alkenones. Rather, this study demonstrates through sequencing that a distinct unique lineage of haptophytes, defined here as Group II-i, produces this compound and the relative contribution of this lineage is greatest in sea ice environments, and shows that this sequence is only found in seawater samples collected under ice-cover. I strongly recommend the publication of this paper for its novelty and importance. I provide several suggestions below for clarification or substantiation of points made in the paper.

The paper combines several lines of evidence, from sequencing of water column and sediment samples and comparison with a wider pool, to determination of the $C_{37:4}$ abundance in culture experiments, surface sediments, and analysis of a sediment time series. The sequencing results and processing are outside of my area of expertise so I will not comment on the methods. From a generalist perspective, the sequencing results for the first time conclusively document the mechanism

behind variations in C37:4 abundance, namely that it is dominated by the relative abundance of a unique haptophyte community.

This sequencing data, and the culture data are important because the normal covariation of many environmental variables spatially makes it difficult to attribute spatial variation in any indicator to a particular environmental variable. In this case, the distribution of C37:4 in sediments serves to confirm the findings from sequencing and culture experiments.

One of the main results for follow up will be the significance of the Group I production of C37:4. Lines 166-167 need some greater specification - what range is meant by "low salinity" and in future studies, what checks should be made in marine sediment sequences to verify that the C37:4 signal is robustly interpreted as changes in sea ice and the significance of the Group II-i community?

The section on the potential role of Group II-i in causing the Uk37 warm bias in polar regions is very speculative and could be supported with some further data (lines 174 to 188). For example, the authors present cultures of a Group II-i strain at constant temperature, in which they measured all alkenone distribution (C37:2 and C37:3) in order to attain the C37:4 relative abundance. For these samples could the Uk'37 ratio be computed to show how it would contribute to a temperature overestimate compared to a calibration from a normal marine species? Presumably if 5-10% C37:4 are a threshold above which Uk'37 temperatures are no longer accurate in marine sediments (As cited here from Rosell et al), then in a pure culture with 80% C37:4, the 37:2 and 37:3 abundance must be very significantly different from that of a normal marine strain at that temperature, and therefore evidence to support this section. Even though only a single temperature is cultured, it is still useful to compare to the Uk'37 at this temperature in normal marine alkenone producers on which the standard calibration is based.

The paragraph from 195-225 presents the evidence that C37:4 abundance in arctic core provides a picture of sea ice extent coherent with other sediment indicators, such as biomarkers from sea-ice diatoms and the coarse fraction, and provides a comparison with temperature. This paragraph would be more effective if at the onset the interpretation of the other proxies were discussed. IP25 from sea-ice diatoms is simple to cite, but the coarse fraction is mentioned in line 204 but only in line 220 is the coarse fraction at this site described to be related to sea ice via brine water rejection and cascades down the shelf. Likewise clarification of the relationship between sea ice and the temperature in east Greenland would be important - is Sea ice a consequence of cold temperatures in east Greenland and therefore the comparison examines forcing and response? Overall, this comparison downcore is interesting and key to the paper but this paragraph has not yet reached optimal clarity.

Response to Reviewers

Reviewer #1 (Remarks to the Author):

What are the noteworthy results?

Wang et al., produce new DNA, culturing and down-core evidence which resolves a long-standing issue in palaeoceanography. Specifically they find evidence that a distinct algal lineage is responsible for the high %C37:4 values observed in POM and sediment samples from sites influenced by Arctic water masses and propose this parameter can be used as a sea-ice proxy.

- Will the work be of significance to the field and related fields? How does it compare to the established literature? If the work is not original, please provide relevant references.
The work is within the context of a well-established suite of biomarkers – alkenones. It's rare for a major step forward in such circumstances, but because of the combination of culturing, next-gen sequencing and new sediment core analyses the authors have achieved this.

- Does the work support the conclusions and claims, or is additional evidence needed?

Yes

- Are there any flaws in the data analysis, interpretation and conclusions? - Do these prohibit publication or require revision?

No – it's a very thorough piece of work.

- Is the methodology sound? Does the work meet the expected standards in your field?

Yes

- Is there enough detail provided in the methods for the work to be reproduced?

Yes

Response: We very much appreciate the positive comments from this reviewer.

Further comments:

The authors are to be commended on a very thorough piece of work. My only critique is that in the set-up and introduction of the paper they somewhat overstate that the community was interpreting %C37:4 as a SSS proxy and that the community had overlooked the possibility that elevated %C37:4 was due to an unidentified producer. In fact workers already suspected this and said as much. For example,

Even as early as 1998, Rosell-Mele was stating “Such large variability in 37:4% must relate to distinct environmental conditions between both regions, in relation to a different 37:4% vs SST relationship perhaps affected by an additional oceanographic parameter and/or interhemispheric species/strains variability.”

And more clearly Bendle et al. 2005 (Wang et. al's ref 22) rejected %C37:4 as a SSS proxy and explicitly stated:

“Our results show that percent C37:4 can be used to reconstruct the relative extension of arctic/polar water masses in the North Atlantic...the results prevent confirmation of percent C37:4 as a paleo-SSS... The fact that *E. huxleyi* is rare in polar waters and *Gephyrocapsa* species are absent in those waters suggests that in this region there might be unknown alkenone

producers associated with polar waters.”

This interpretation was used by McClymont et al., 2008 and others who were careful not to interpret %C_{37:4} as a SSS proxy.

The authors should amend their introduction and discussion to reflect this prior thinking. For example at line 81-83 they state “Such conflict suggests that elevated %C_{37:4} is not a physiological response to decreased SSS, but rather that unidentified producers of C_{37:4} occur in high latitude marine environments”.

Without citing any previous work (examples above) they give the reader the misleading impression that they are the first workers to propose this idea. They are not, but they are the first to prove it (for which they are commended)! They need to make a clear distinction in the introduction and discussion that are proving an idea that was already proposed by others.

Response: We agree with the reviewer comments and have made following revisions (line 74-79):

The mechanisms for these observations, however, are poorly defined, with previous studies suggesting unknown oceanographic parameters¹⁴, and/or alkenone producers²² as the possible explanations. Miscellaneous compounds co-eluting with C_{37:4} alkenone from sediment samples add an additional level of complication for interpreting %C_{37:4} data.²³ Regardless of these problems, elevated %C_{37:4} is widely applied as an indicator of fresher and colder polar water mass or decreased SSS (Supplementary Table 1).

We have added reference of Rosell-Melé 1998, and Bendle et al., 2005 at line 85.

Table S1: Need to make clear that these papers refer to down-core records (and not POM). Change from “List of published %C_{37:4} records...” to “List of published %C_{37:4} sediment-core records...”

Response: Changed as suggested.

Reviewer #2 (Remarks to the Author):

This is a well written and well executed study. The discovery of an extensive new lineage of C_{37:4} alkenone producing group II isochrysidales and the discovery that %C_{37:4} can be used as a proxy to reconstruct sea ice extension in high latitude oceans for possibly millions of years is well worth publishing in Nature Communications. The study focused on the Arctic and it remains to be seen if the proxy also work for Antarctic environments. Namely, the newly discovered lineage is distantly related to the C_{37:4} mK producers previously recovered from Ace Lake in the Vestfold Hills, Antarctica. These sequences were included in the phylogenetic tree but are clearly clustering in a separate group.

Response: We appreciate the positive comments from this reviewer.

I only found a minor issue in the wording of the methods between lines 280-284. The sentence in line 280 ", except studies that quality trimmed ASVs were available" does not run well. IN line 282 the word "the" is missing after "using". The choice for the stand-alone sentence starting with "Sequences classified to the order... in line 284 is a bit odd. I am not sure what this refers to in the context that it is written.

Response: We have made following edits:

Line 295: "except for studies with available quality trimmed ASVs^{40,64}."

Line 297: "identified using the assignTaxonomy command"

Line 299-300: "Sequences classified to the order Isochrysidales were selected to proceed to further analysis."

Reviewer #3 (Remarks to the Author):

The study of polar climate change is a very timely topic given the vulnerability of polar regions to anthropogenic change. A new tool for assessing the past coverage of sea ice, proposed here, is especially key because climate models increasingly invoke sea ice as a major amplifier of climate perturbations from volcanic cooling to shutdown of Atlantic meridional overturning circulation. Therefore the paper is likely to be of very high interest to a broad research community, and is likely to catalyze both new studies and a wide re-interpretation of many existing datasets.

The article provides an important and compelling new interpretation for the contribution of an unusual lipid biomarker in sediments and thereby a new perspective on sea ice and polar climate change. For decades the concentration of C37:4 alkenone has been observed to increase at certain periods in North Atlantic cores, but the environmental meaning of this change has been much disputed, with interpretations ranging from extreme cold to extreme low salinity. This study convincingly shows through laboratory culture experiments isolating a C37:4 producing strain, that salinity does not affect the C37:4 ratio of the produced alkenones. Rather, this study demonstrates through sequencing that a distinct unique lineage of haptophytes, defined here as Group II-i, produces this compound and the relative contribution of this lineage is greatest in sea ice environments, and shows that this sequence is only found in seawater samples collected under ice-cover. I strongly recommend the publication of this paper for its novelty and importance. I provide several suggestions below for clarification or substantiation of points made in the paper.

Response: We appreciate the positive comments from this reviewer.

The paper combines several lines of evidence, from sequencing of water column and sediment samples and comparison with a wider pool, to determination of the C37:4 abundance in culture experiments, surface sediments, and analysis of a sediment time series. The sequencing results and processing are outside of my area of expertise so I will not comment on the methods. From a generalist perspective, the sequencing results for the first time conclusively document the

mechanism behind variations in C37:4 abundance, namely that it is dominated by the relative abundance of a unique haptophyte community.

This sequencing data, and the culture data are important because the normal covariation of many environmental variables spatially makes it difficult to attribute spatial variation in any indicator to a particular environmental variable. In this case, the distribution of C37:4 in sediments serves to confirm the findings from sequencing and culture experiments.

One of the main results for follow up will be the significance of the Group I production of C37:4. Lines 166-167 need some greater specification - what range is meant by "low salinity" and in future studies, what checks should be made in marine sediment sequences to verify that the C37:4 signal is robustly interpreted as changes in sea ice and the significance of the Group II-i community?

Response: We have made clarifications and added relevant citations as suggest:

Line 168: We have added "0-6 ppt" to specify the meaning of "low salinity". This range is previously defined in Long et al. (2016), reference 39.

Line 170-172: We also added a sentence to clarify the detection of Group 1: "Alkenone contribution from Group 1 can be identified through the presence of C_{37:3b} isomer, which is not produced by Group 2 and 3⁴¹."

The section on the potential role of Group II-i in causing the Uk37 warm bias in polar regions is very speculative and could be supported with some further data (lines 174 to 188). For example, the authors present cultures of a Group II-i strain at constant temperature, in which they measured all alkenone distribution (C37:2 and C37:3) in order to attain the C37:4 relative abundance. For these samples could the Uk'37 ratio be computed to show how it would contribute to a temperature overestimate compared to a calibration from a normal marine species? Presumably if 5-10% C37:4 are a threshold above which Uk'37 temperatures are no longer accurate in marine sediments (As cited here from Rosell et al), then in a pure culture with 80% C37:4, the 37:2 and 37:3 abundance must be very significantly different from that of a normal marine strain at that temperature, and therefore evidence to support this section. Even though only a single temperature is cultured, it is still useful to compare to the Uk'37 at this temperature in normal marine alkenone producers on which the standard calibration is based.

Response: We agree with the reviewer, which is why we have already written the manuscript in cautious tones (e.g., line 182-183: however the exact mechanism leading to the warm bias is still a question) and with lots of detailed discussions. In the discussion text from line 186 to 192, We have listed a range of complicated factors that can ultimately cause the bias of UK37' inferred temperature. To further address the reviewer concern, we added following sentence to the end of the paragraph (line 192-194): "More experimental and field studies are needed to fully understand the causes for the warm SST bias inferred by UK37' in high latitude oceans."

The paragraph from 195-225 presents the evidence that C37:4 abundance in arctic core provides

a picture of sea ice extent coherent with other sediment indicators, such as biomarkers from sea-ice diatoms and the coarse fraction, and provides a comparison with temperature. This paragraph would be more effective if at the onset the interpretation of the other proxies were discussed. IP₂₅ from sea-ice diatoms is simple to cite, but the coarse fraction is mentioned in line 204 but only in line 220 is the coarse fraction at this site described to be related to sea ice via brine water rejection and cascades down the shelf. Likewise clarification of the relationship between sea ice and the temperature in east Greenland would be important - is Sea ice a consequence of cold temperatures in east Greenland and therefore the comparison examines forcing and response? Overall, this comparison downcore is interesting and key to the paper but this paragraph has not yet reached optimal clarity.

Response: We have re-arranged the relevant sentences and references, so that the significance of each proxies (e.g, IP₂₅ and coarse fraction) is promptly explained. We also clarified the relationship between sea ice and Greenland temperature.

Line 204-208: “IP₂₅ is a C₂₅ isoprenoid lipid synthesized by certain Arctic sea ice diatoms and is deposited in underlying sediments following ice melt in spring⁵³⁻⁵⁵. Coarse sediment fraction at this site is largely composed of lithic grains and foraminifera, reflecting sea ice formation off the western Barents shelf⁵⁶.”

Line 208-212: “The Fram Strait is the major gateway between the North Atlantic and the Arctic Ocean, and sea ice distribution in our study area is highly sensitive to surface temperature variability in modern time⁵⁷. Over longer timeframes such as the Holocene, variations in sea-ice extent in the Fram strait, the Greenland Sea and the northern Barents Sea, co-occurred with changes in Northern Hemisphere solar insolation and changes in Greenland air temperature⁵⁸⁻⁶².”